# An Update on the Role of Extracellular Vesicles in the Pathogenesis of Necrotizing Enterocolitis and Inflammatory Bowel Diseases

**DOI:** 10.3390/cells10113202

**Published:** 2021-11-17

**Authors:** Rafał Filip

**Affiliations:** 1Department of Gastroenterology with IBD, Unit of Clinical Hospital 2 in Rzeszow, Lwowska 60, 35-310 Rzeszow, Poland; r.s.filip@wp.pl; 2Faculty of Medicine, University of Rzeszow, Aleja Majora Wacława Kopisto 2a, 35-210 Rzeszow, Poland

**Keywords:** extracellular vesicles, gut maturation, necrotizing enterocolitis, inflammatory bowel disease

## Abstract

Some of the most fundamental influences of microorganisms inhabiting the human intestinal tract are exerted during infant development and impact the maturation of intestinal mucosa and gut immune system. The impact of bacteria on the host gut immune system is partially mediated via released extracellular vesicles (EVs). The heterogeneity in EV content, size, and bacterial species origin can have an impact on intestinal cells, resulting in inflammation and an immune response, or facilitate pathogen entry into the gut wall. In mammals, maintaining the integrity of the gut barrier might also be an evolutionary function of maternal milk EVs. Recently, the usage of EVs has been explored as a novel therapeutic approach in several pathological conditions, including necrotizing enterocolitis (NEC) and inflammatory bowel disease (IBD). In this review, we attempt to summarize the current knowledge of EV biology, followed by a discussion of the role that EVs play in gut maturation and the pathogenesis of NEC and IBD.

## 1. Introduction

Extracellular vesicles (EVs) were originally considered as intracellular residues excreted by cells to dispose of metabolic wastes [1]. However, studies from the last two decades indicate that EVs are in fact a nonclassical means of intercellular communication, along with classical means of cell–cell contact and secretion of various signaling molecules [2,3]. Increased interest in EVs has been observed since they are now considered novel contributors to physiological and pathological processes due to their capability to alter a number of molecular reactions [4].

### 1.1. Biological Role and Components of EVs

These tiny vesicles encapsulate a variety of molecules, including proteins, lipids, sugars, and nucleic acids. They are surrounded by a lipid bilayer membrane and can be easily transferred in the lumen and/or within the intestinal wall without losing biological activity. The lipophilic properties of EVs, apart from protecting against extracellular enzymes and the aqueous environment, facilitate interactions with target cells as they can be easily combined with cell membranes and endocytosed [5]. Because of this, circulating EVs are beginning to be recognized as not only biomarkers, but also as possible regulators in a number of gastrointestinal disturbances. Most cells and tissues are capable of excreting EVs, including hepatocytes, leukocytes, endothelial cells, and epithelial cells. Moreover, they can also function as recipients or effectors of EV targeting. Cell uptake of Evs varies according to cell type, but most commonly occurs as clathrin-mediated endocytosis, caveolin-mediated endocytosis, lipid raft-mediated endocytosis, macropinocytosis, or phagocytosis [2,6]. Extracellular vesicles have also been detected in various body fluids, including plasma, bile, saliva, ascitic fluid, as well as abdominal drainage fluid [7].

The role of EVs is well recognized in many important human pathologies, such as cardiovascular or neurodegenerative diseases [8,9]. Due to their unique modalities, they have also been investigated for use as biomarkers or delivery vehicles in therapeutic applications [10]. Finally, EVs may also have a significant role in aging. Experiments performed in older mice demonstrated that structural and quantitative modifications of EVs can ameliorate age-related functional decline [11,12]. On the other hand, it was also shown that many pathogenic bacteria use their EVs to deliver toxic compounds to infected cells [13]. Bacteria-derived EVs consist of an outer membrane and periplasmatic proteins, lipopolysaccharides (LPS), DNA, RNA, and other factors associated with virulence [14]. Therefore, the role of infectious agents in the etiology of disturbances once believed to be non-infectious, such as insulin resistance, is increasingly being studied [15].

### 1.2. Classification of EVs

Because new isolation methods for EV subtypes are constantly being developed, a clear consensus on EV nomenclature has not yet been established. Therefore, a number of prerequisites have been proposed by the International Society for Extracellular Vesicles (ISEV) to confirm the presence of EVs in analyzed samples [16]. Taking into account particle size, release from cells, and mechanism of formation, a distinction between exosomes, microvesicles (MVs), and apoptotic bodies (ABs) has been proposed in eukaryotes. However, the stipulation that EVs could vary within a group, depending on the type of cargo and functionality, was also included. Exosomes arise as intraluminal vesicles (ILVs) within a multivesicular body (MVB), which are released into the extracellular matrix after association with the plasma membrane [17]. Microvesicles (ectosomes) are 200–1000 nm microparticles originating from the plasma membrane and are subsequently released via actomyosin-driven fission of plasma membrane vesicles. Microvesicles extracted from cancer cells, also called “oncosomes” (1–10 µm), are involved in the horizontal transfer of signaling molecules contributing to cancer invasiveness [18,19]. Apoptotic bodies are larger, include vesicles (1–5 µm), and originate from membrane blebbing and cellular disassembly from cell fragmentation when the cytoskeleton disrupts during the beginning of apoptosis. Although the biological function of ABs has not yet been fully elucidated, their role in immune process regulation, including autoimmunity and carcinogenesis, has been described [20]. The presented classification does not refer to bacteria-released membrane vesicles (20–400 nm), which are regarded as MVs or outer membrane vesicles (OMVs), both of which contain DNA, sRNA, proteins, and other biologically active molecules. However, OMVs refer to those arising from Gram-negative bacteria with a diameter of approximately 20–400 nm, while MVs are cytoplasmic membranes of Gram-positive bacteria with a typical diameter of 20–150 nm [21].

Outer membrane vesicles are considered to represent a specialized bacterial secretion pathway, however the differences in bacterial physiology imply different pathways of EV release, which in turn may lead to the differences in types of EVs. In Gram-negative bacteria, EVs are released when the outer membrane lipid asymmetry is violated, causing membrane curvatures, which lead to membrane vesiculation. This process may be triggered by a decrease in protein linkages between the outer membrane and peptidoglycans, or a change in shape of transmembrane proteins. Subsequently, accumulation of unfolded proteins in the periplasmatic space leads to release of EVs through the exertion of turgor pressure [22]. Although many studies have been carried out in this field, the mechanisms underlying the biogenesis of gram-positive bacterial EVs has not yet been fully elucidated. However, it was shown that endolysin triggers “bubbling cell death”, leading to the release of cytoplasmic membrane vesicles (CMVs), which can contain both cytoplasmic and cell membrane components [23,24].

Although there have been many recent advances in the field of EV research, many questions regarding the function of EVs in homeostasis or disease remain unanswered [25]. Enterocytes (EC) provide a semipermeable barrier limiting the amount of antigen reaching the epithelial surface, which forms the basis of endothelial integrity. The resistance of the tight junctions (TJs) regulates the paracellular transport of antigens from the gut lumen. The integrity of this barrier is an important determinant of gut homeostasis, and intestinal hyperpermeability is a hallmark of the inflammatory response to injury or infection [26].

### 1.3. Possible Role for Extracellular Vesicles in Gut Maturation

In infants, microbial colonization after birth represents a critical period in terms of gastrointestinal system development. Disturbances in the acquisition of healthy microbiota may lead to acute or chronic pathologies such as necrotizing enterocolitis (NEC), sepsis, as well as a number of complex lifestyle-related and age-related disorders, such as metabolic, neurodegenerative, and inflammatory diseases [27,28,29,30]. Increasing evidence also suggests that under the influence of gut microbiota, the risk for acquiring diseases may be programmed during the fetal developmental period and early life [31,32]. Some of the most fundamental influences of microorganisms inhabiting the human intestinal tract are exerted during infant development and impact the maturation of intestinal mucosa and gut immune system [33].

#### 1.3.1. EVs of Bacterial Origin

The impact of bacteria on the host gut immune system is mediated via released EVs. The heterogeneity in EV content, size, and bacterial species origin can have an impact on intestinal cells, resulting in inflammation and an immune response, or facilitate pathogen entry into the gut wall. Bacterial EVs contain a variety of cell surface components, including outer membrane proteins (OMPs), peptidoglycan (PG), and LPS, and may also function as transporters of different bacterial components such as nucleic acids, enzymes, toxins, and a complex of microbe-associated molecular patterns (MAMPs). Therefore, a number of enterocyte pattern recognition receptors (PRR), such as Toll-like receptors (TLRs) and Nod-like receptors (NLRs), may be triggered by different EVs [34]. Recent data have clearly demonstrated that the composition of gut microbiota may significantly influence the gene expression of intestinal tissue. Furthermore, it was shown that the bacterial RNA found in OMVs aligned to some of the host chromosomes, which suggests that OMVs might be able to affect epigenetic processes [34,35,36].

On the other hand, it was also demonstrated that EVs are one of the key mediators of paracrine signaling, such as between stem cells and target tissues, and if concentrated, can be greater than that of the parent cells [37,38,39].

#### 1.3.2. EVs of Immune System Origin

Similar to observations in bacteria, stem cell EVs exert their role as intercellular communicators by providing their “cargo”, containing selected lipids, proteins, as well as nucleic acids [16]. It is important to note that EVs transporting RNA particles are capable of epigenetically regulating target cell genes [40]. Therefore, from a clinical application perspective, EVs are being considered as substitutes for stem cell-based therapy in tissue regeneration, including in NEC [39].

Necrotizing enterocolitis is a common and devastating gastrointestinal condition of the immature gut, predominantly affecting premature infants. The etiology of NEC has yet to be fully elucidated. Inflammatory processes within the gut wall may vary from mucosal injury to full-thickness necrosis and perforation, which may cause systemic inflammation affecting distant organs, such as the brain. Apart from mucosal immaturity, hypoxia, and disturbances in the intestinal microflora are also considered to be main factors in its pathogenesis. Thus far, the most effective interventions have been preventative approaches, in particular, a consistent feeding protocol, slow initiation and advancement of feeds, the use of human milk, prophylactic antibiotics, and probiotics [41].

However, recent studies into the usage of EVs as a novel therapy have been promising. In animal models, it was shown that the biologically active vectors of bone marrow-derived mesenchymal stem cells (BM-MSCs) administered to neonates with NEC were the exosomes excreted by the cells [42]. These exosomes were capable of restoring intestinal barrier function, similar to an infusion of BM-MSCs alone. Moreover, it was observed that intraperitoneal infusion of BM-MSC exosomes may lead to decreased severity of intestinal injury and overall incidence of NEC. A study by Rager et al. clearly showed that stem-cell-derived EVs play a significant role in paracrine signaling, and that it is possible to use EVs as a cell-free therapy for NEC in newborns [42]. Another study using a rat model showed that amniotic-fluid-derived stem cells (AF-MSC), amniotic-fluid-derived neural stem cells (AF-NSC), and neonatal enteric NSC (E-NSC)—derived exosomes are capable of significantly reducing the incidence of experimentally induced NEC [43]. Moreover, these exosomes appeared to be as therapeutically effective as the stem cells from which they were derived [43]. A similar beneficial paracrine action of various stem cells was also seen in animal models of bladder, heart, kidney, and lung diseases [44]. Although these results in animal models are very promising, the molecular mechanisms through which EVs specifically induce the maturation of intestinal mucosa have not yet been clearly elucidated. In a study by Good et al., amniotic fluid was shown to decrease the severity of intestinal damage in experimental NEC through inhibition of the TLR4 signaling pathway [45]. Using selective and non-selective cyclooxygenase 2 inhibitors in survival studies, other researchers have shown that the beneficial effect can be achieved via modulation of stromal cells expressing cyclooxygenase 2 in the lamina propria. Interestingly, AF-MSC cells differentially expressed genes of the Wnt/β-catenin pathway, which regulate intestinal epithelial stem cell function and cell migration, along with growth factors which are known to control the integrity of intestinal epithelium and abate mucosal injury. This led to a significantly decreased incidence of NEC, as well as decreased macroscopic gut damage, improved intestinal function, increased enterocyte proliferation, and reduced apoptosis [46]. Other studies have confirmed the paracrine action of AF-MSCs by showing that the effect imposed by AF-MSCs on the impaired neonatal gut was present despite a low degree of cell engraftment. Additionally, released molecules capable of regenerating immature or impaired intestine may be released by MCSs, independent of their origin, such as from the umbilical cord or bone marrow [43,47].

#### 1.3.3. EVs of Maternal Milk Origin

The crucial role of maternal milk in the development and maturation of the GI tract, as well as in the maintenance of immune system homeostasis is well recognized; however, the exact molecular pathways remain unclear. Recent studies indicate that purified EVs from human milk can induce gingival cell re-epithelialization, modify epithelial endosomal TLR responses, and transiently control T-cell activation [48]. More specifically, Zonneveld et al. showed that EVs derived from maternal milk positively influence epithelial barrier function by enhancing cell migration via the p38 MAPK pathway. In addition, maternal milk EVs downregulate activation of endosomal TLR3 and TLR9. Researchers have also demonstrated that maternal milk EV signaling molecules are capable of modulating cellular processes, not only in various GI tract cells, but can also directly inhibit activation of CD4+ T-cells by deactivating T-cells without inducing tolerance [48]. Apart from enterocytes and the gut immune system, other cells composing the intestinal wall can also benefit from the trophic activity of EVs, such as maternal milk exosomes, which are capable of enhancing mucin production and increasing the expression of goblet cell-associated markers (trefoil factor 3 and mucin 2) [49].

Based on studies using bovine, porcine, and rat models [50,51], it is reasonable to conclude that maintenance of the gut barrier integrity might be an evolutional function of maternal milk EVs in mammals [52]. Some genetic studies support this hypothesis, which have demonstrated the presence of EV miRNA cargo with regulatory functions in maternal milk that are conserved among various mammalian species [52]. A study by Martin et al. demonstrated that human breast milk exosomes prevent cell death in intestinal epithelial cells (IECs) by guarding against H_2_O_2_-induced oxidative stress [53]. This is consistent with data from Chen et al., which showed that porcine milk-derived exosomes positively influence gut maturation indices such as intestinal mucosa villus height and crypt depth [54]. Surprisingly, it was shown that preterm exosomes are more effective when compared to their full-term counterparts with respect to the stimulatory effect on epithelial cell proliferation [55].

In this review, we attempt to summarize the current knowledge of EV biology, followed by a discussion of the role that EVs play in gut maturation and the pathogenesis of IBDs.

## 2. Extracellular Vesicles in the Pathogenesis of IBD

### 2.1. The Role of EVs in Maintaining the Gut Integrity

Increased intestinal barrier permeability is considered to be the result of mucosal inflammation, and restoration of the intestinal barrier is a key component in maintaining intestinal homeostasis during IBD quiescence.

Within enterocytes, proper homeostasis in the digestive system depends on the interplay between intracellular, extracellular, and intercellular communication pathways. Direct intercellular communication is possible via gap junctions, composed of two hemichannels of neighboring cells, which govern the diffusion of specific molecules between adjacent cells. This phenomenon is known as gap junctional intercellular communication (GJIC), and involves number of second messengers, including adenosine triphosphate (ATP), cyclic adenosine monophosphate (cAMP) and inositol triphosphate (IP3) [56]. The intestinal lumen contains a very complex environment with numerous microbiome species, and its equilibrium is based on a dynamic and sophisticated interplay between the epithelial layer, gut immune system, and microbiota. Control is achieved through miRNA, regulatory molecules, and receptors, as well as a number of signaling pathways which allow the maintenance of intraluminal symbiosis and prevents overexpression of the host immune response. Recent studies have demonstrated that indirect communication based on the mutual interplay between bacterial and host EVs are essential in maintaining gut lumen homeostasis [57]. Moreover, the disease state in the gut can be reflected by specific proportions and amounts of host-derived EVs, such as in IBD patients, which indicate that those EVs may have an effect on the inflammatory state [58]. Immunological processes can alter EV content, increasing the amount of cyto- and chemokines, adhesion molecules, growth factors, and matrix metalloproteinases (MMPs) [58]. Cell-derived EVs and microbiota-derived EVs add to intestinal barrier integrity and can significantly influence gut immune system function. Their actions are driven by their specific cargo, which can influence target cells, such as via membrane receptors. Importantly, there is increasing evidence which indicates that altered intra- or extraluminal conditions, such as in inflammation, hypoxia, or extreme pH, could determine the content and amount of EVs [59]. Taken together, it appears that molecules carried by EVs, such as transforming growth factor β1 (TGF-β1) and annexin-1 (ANXA1), are associated with specific pathologies, including those of gut barrier integrity maintenance in IBD [60,61] (Figure 1 and Table 1).

#### 2.1.1. Transforming Growth Factor β (TGF-β)

TGF-β is a multifunctional growth factor which promotes healing of the overlying epithelium by modulating epithelial cell restoration, which serves to re-establish surface continuity after mucosal injury. Previous studies have shown that loss of TGF-β signaling increases susceptibility to ulcerative colitis (UC) [60,62]. However, more recent studies have elucidated the mode of action by which TGF-β contributes to restoration of immune balance in the intestinal tract and have also explained how TGF-β is delivered to target cells. Jiang et al. demonstrated that under physiological conditions, EVs with TGF-β1-dependent immunosuppressive activity are released by IECs. Transfer of these EVs into mice with experimentally induced IBD decreased colon shortening, reduced body weight loss, and lowered inflammation indices through the induction of regulatory T-cells and immunosuppressive dendritic cells (DCs). Moreover, localization of these EVs was associated with epithelial cell adhesion molecule (EpCAM). In mice IBD models, knockdown of EpCAM led to increased inflammatory changes. Additionally, the protective effect of EVs from IECs with decreased EpCAM on murine IBD was less visible [63].

#### 2.1.2. Annexin-1 (ANXA1)

ANXA1 is an inflammatory modulator and is a potential link between systemic inflammation and gut immune dysfunction in the course of IBD. This calcium-dependent phospholipid binding protein is expressed by intestinal epithelium and acts by binding to formyl peptide receptors (FPRs). Babbin et al. have demonstrated increased susceptibility to chemically induced mucosal inflammation in mice lacking ANXA1. In addition, ANXA1-deficient mice exhibited impaired clinical and histopathologic recovery following an inflammatory state [64]. Other researchers have shown ANXA1 to be released as a component of EVs derived from intestinal epithelium, and these ANXA1-containing EVs activate wound repair circuits. Moreover, it was shown that patients with active IBD tended to have elevated levels of secreted ANXA1-containing EVs in their sera. Since ANXA1-containing EVs are systemically distributed in response to the inflammatory process, they can potentially be used as a biomarker of intestinal inflammation [65]. This hypothesis was also supported by Paula-Silva et al., who demonstrated that ANXA1 contributes to the restoration of intestinal homeostasis after TNF-α inhibition with infliximab (used in the treatment of IBD), constituting a potential biomarker of therapeutic efficacy [66]. However, there are also available studies indicating that the ANXA1/EVs-FPR axis may be involved in cancer progression as a mediating vehicle of cell–stroma intercommunication [67].

#### 2.1.3. MicroRNAs (miRNAs)

MiRNAs are noncoding RNA fragments that regulate gene expression and can be encapsulated in EVs (exosomes). They are selectively and actively loaded into EVs and then transferred to the target recipient cell where they manipulate cell function through post-transcriptional modifications. MiRNA contained in EVs differs from their cellular counterparts, which indicate an active sorting and packaging mechanism of EV miRNAs. These have recently gained substantial interest because of their importance in several biological processes, such as cell cycle control, cell differentiation, and apoptosis. By regulating cytokine expression, miRNAs are involved in inflammation, and therefore, might play an important role in the pathogenesis of many autoimmune and inflammatory diseases [68]. The molecular pathways of miRNAs have not yet been fully elucidated; however, it is known that they can act as agonists of the single-stranded RNA-binding TLRs [69]. Via TLR signaling, activation of NF-κB leads to the secretion of a number of molecules with pro-inflammatory properties, which indicates that miRNAs can enhance inflammatory processes. MiRNA-21 and miR-29a encapsulated in extracellular EVs can increase secretion of TNF and IL 6 by binding to murine TLR7 and to TLR8 in human macrophages [70]. Moreover, it was shown that a discrete pool of miRNAs, which can reprogram endothelial cells upon internalization, can be found in platelet-derived microvesicles [71]. This could be important in the understanding of diseases with an inflammatory origin. In addition to the previously mentioned pathways with miRNAs released in EVs, there are other mechanisms and pathways which are involved in the maintenance of gut homeostasis via different target genes and pathways triggered by miRNAs transported by GI-tract-derived EVs [72]. One example is micro-RNA-21 (miR-21), which influences epithelial integrity through modification of the PTEN/PI3K/Akt signaling pathway and targeting Ras-related small GTP-binding protein B (RhoB) and protein 42 (CDC42), which are responsible for cell division. This pathway is of great interest, since it was shown that in IBD patients, serum levels of miR-21 are elevated [73,74]. Another exosomal miRNA, miR-29, has been shown to modify pro-inflammatory cytokine release and scavenger receptor expression by targeting LPL in ox LDL-stimulated DCs [75]. A study by Lv et al. also demonstrated that miR-29a is highly expressed in the colonic tissues of UC patients and mice with dextran sulfate sodium (DSS)-induced experimental colitis. Moreover, they showed that miR-29a targeted the 3′UTR of the *Mcl-1* gene and downregulated the expression of Mcl-1 in colonic tissues in the experimental murine model of colitis as well as in UC patients [76]. Among other miRNAs, miR-223 has also gained interest since it was demonstrated that in a 2,4,6-trinitrobenzene sulfonic acid (TNBS)-induced colitis mice model, miR-223 acts as a mediator of the IL-23 pathway which suppresses claudin-8. It is well recognized that the IL-23 pathway plays an important role in IBD pathogenesis due to the downregulation of claudin-8, a key protein that conforms to the structure of intestinal TJs [77]. Furthermore, a study from 2020 by Li et al. showed that exosomes from human mast cells contain significant amounts of miR-223 and that acquisition of miR-223 into IECs downregulates the expression of TJ-related proteins ZO-1, occludin, and claudin-8, which may lead to a subsequent increase in intestinal permeability and inflammatory cascade [78].

#### 2.1.4. Human Proteins Related to Oxidative Antimicrobial Activities

In 2016, Mottavea et al. used mucosal–luminal interface aspirates to demonstrate the important roles of mitochondrial dysfunction and aberrant host-microbiota crosstalk in newly diagnosed Crohn’s disease (CD). It is important to note here that this experiment highlighted the value of direct sampling at the site of disease, primarily because these interactions were mediated by proteins encapsulated in EVs [79]. Two years later, Zhang et al. performed the first proteomic characterization of intestinal EVs and demonstrated for the first time that host defense proteins, including those which produce reactive oxidants, were present in the intestinal EVs and correlated with functional alterations in the microbiota of pediatric IBD patients [80].

**Figure 1 cells-10-03202-f001:**
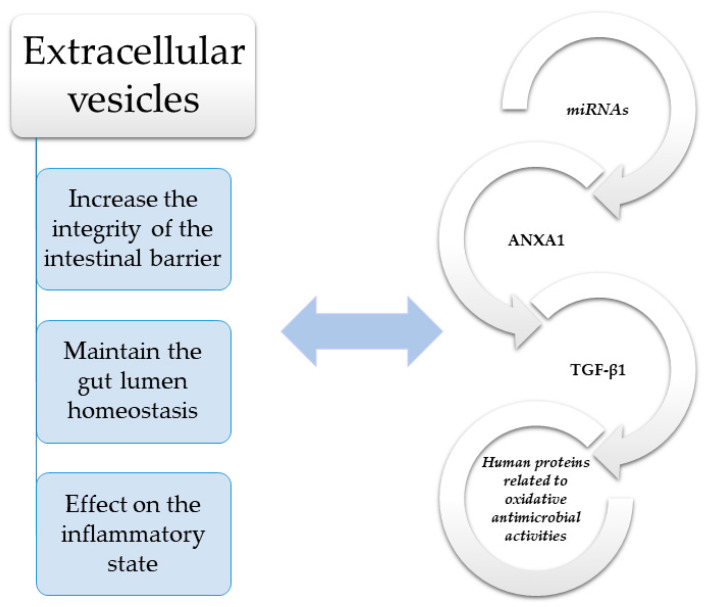
Molecules such as transforming growth factor β1 (TGF-β1) and annexin-1 (ANXA1) have a significant impact on maintaining the integrity of the intestinal barrier in IBD [60,61]. TGF-β—Transforming growth factor β; ANXA1—Annexin-1; miRNAs—MicroRNAs.

**Table 1 cells-10-03202-t001:** Extracellular (EVs) vesicles in the pathogenesis of IBD.

Extracellular Vesicles	Role in the Pathogenesis of IBD
TGF-β	• EVs with TGF-β1-dependent immunosuppressive activity are released by IECs [63].
	• EVs with TGF-β1-dependent may lead to reduction inflammation indices through the induction of regulatory T-cells and immunosuppressive dendritic cells [63].
ANXA1	• ANXA1-deficiency may exhibit impaired clinical and histopathologic recovery following an inflammatory state [64].
	• ANXA1 may be released as a component of EVs derived from intestinal epithelium and it may activate wound repair circuits [65].
	• ANXA1-containing EVs can potentially be used as a biomarker of intestinal inflammation [65].
miRNAs	• MiRNA contained in EVs may affect cell cycle control, cell differentiation and apoptosis [68].
	• MiRNA can act as agonists of the single-stranded RNA-binding TLRs [69].
	• MiRNAs can enhance inflammatory processes (increase secretion of TNF and IL 6) [70].
	• The maintenance of gut homeostasis via modification of the PTEN/PI3K/Akt signaling pathway [73,74].
	• MiR-29 has been shown to modify pro-inflammatory cytokine release and scavenger-receptor expression by targeting LPL in ox LDL-stimulated DCs [75]
	• MiR-223 can act as a mediator of the IL923 pathway which suppresses claudin-8 [74].
	• Acquisition of miR-223 into IECs downregulates the expression of TJ-related proteins ZO-1, occludin, and claudin-8 [78].
Human proteins related to oxidative antimicrobial activities	• Host defense proteins, including those which produce reactive oxidants are correlated with functional alterations in the microbiota of pediatric IBD patients [80].

TGF-β—Transforming growth factor β; ANXA1—Annexin-1; miRNAs—MicroRNAs; IECs—Intestinal epithelial cells, DC-dendric cells.

### 2.2. The Role of EVs in the Regulation of the Gut Immune System

#### 2.2.1. EVs of Intestinal Epithelial Cell (IEC) Origin

Intestinal epithelial cell-derived EVs contain a variety of molecules which take part in crosstalk with the immune cells, and therefore, are considered key messengers for intercellular communication in the gut [81,82]. The target and function of IEC-derived EVs depends on the type of cargo that they carry. For example, under physiological conditions, TGF-β1 is one its main components. Jiang et al. demonstrated in a murine model of IBD that IEC-derived EVs containing TGF-β1 may decrease inflammation by stimulating immunosuppressive DCs and Treg cells. These vesicles mainly localized with the EpCAM within the gut wall. The protective effects of TGF-β1-rich EVs are dampened in EpCAM knockout mice, which indicates that EpCAM-dependent EVs from IECs are important players in maintaining intestinal tract immune balance [63]. On the other hand, there is growing evidence that EVs derived from the basolateral side of enterocytes are independent antigen-carrying structures actively participating in the presentation of luminal antigens to the intestinal immune system. Studies of samples taken from the small intestine revealed that IECs express MHC I and MHC II, HLA-DM, CD63, CD68, A33, and other molecules normally found in exosomes derived from typical antigen-presenting cells [83]. Intestinal epithelial cell-derived EVs are capable of participating in antigen presentation within the intestinal mucosa, independently of direct interaction with effector cells. This was demonstrated in the murine epithelial cell line MODE K. This study revealed that the excreted EVs display MHC I/II peptides and other immune markers, such as CD9, CD81, CD82, and A33 antigen and were able to activate immune responses depending on the exosome-expressed epitopes [84]. This was also confirmed in a study by Mallegol et al., who demonstrated that direct communication between IEC line T-84 and DCs via exosomes leads to stronger antigen presentation, which in turn may lead to effector T-cell response [85].

#### 2.2.2. EVs of Immune System Cell Origin

It is currently accepted that both the innate and adaptive immune responses contribute to IBD development, and the strong dependency between abnormal immune system functioning and IBD is well recognized. Although IECs are key players in cell-to-cell crosstalk within the GI tract, the proper immunological response to pathogens is also dependent on transmission of information to the underlying immune cells via the release of EVs and soluble mediators. This pertains to most immune cells, including neutrophils, macrophages, monocytes, and DCs. The effector T-cell response is triggered by DCs through antigen presentation as well as the release of EVs containing MHC class I and II complexes and other mediators which contribute to antigen presentation and immunomodulatory effects toward CD4+ or CD8+ T-cells. In a study from 2002, Théry et al. showed that EVs extracted from DCs induce antigen-specific naïve CD4+ T-cell activation. Moreover, they also demonstrated in cell cultures that only EVs derived from mature DCs expressing CD80 and CD86 costimulatory markers could trigger naïve CD4+ activation [86]. Further studies revealed that DCs previously exposed to LPS release exosomes containing MHC class II, B7.2, and intercellular adhesion molecule 1 (ICAM-1) [87]. These observations are important from the perspective of IBD pathogenesis, and during the last few years, much progress has been made in understanding the role of immune-cell-derived EVs. Cells other than DCs can also participate in GI immune system crosstalk. Because of this, it has been suggested that Evs might share some properties with their parent cells.

Wang et al. demonstrated that that granulocytic myeloid-derived suppressor cell (G-MDSC) exosomes are capable of inhibiting CD4+ T-cell proliferation and IFN-γ secretion, and that these observations correlated with Arg-1 activity. Additionally, they found that G-MDSC EVs promoted Tregs expansion from CD4+ T-cells in the presence of TGF-β. Taken together, these results confirm the strong immunosuppressive properties of G-MDSC-derived EVs [88]. On the other hand, in a study from 2016, Wong et al. provided new insight into the role of macrophage-derived EVs in the pathogenesis of IBD. They incubated RAW264.7 macrophages with serum exosomes extracted from DSS-induced mice and demonstrated that treatment may induce phosphorylation of p38 and ERK and the release of TNF-α when compared to incubation with exosomes isolated from control mice. This in vitro experiment confirmed the capability of DSS EVs in activating the MAPK signaling pathway and triggering a pro-inflammatory response in naïve macrophages. Furthermore, proteomic characterization of those EVs revealed 56 differentially expressed molecules, with acute phase proteins and immunoglobulins among them [89]. Polymorphonuclear leukocyte (PMN) infiltration of the intestinal mucosa usually leads to damage of the epithelial layer. Under inflammatory conditions, PMNs, in parallel with the production of exosomes having direct antimicrobial properties, can also release EVs which interfere with the maturation of monocyte-derived DCs. Eken et al. have shown that stimulation of immature DCs with PMN-derived EVs enhances TGF-β1 release, downregulates phagocytic activity of DCs, and induces changes in DC morphology. Furthermore, incubation of immature DCs with PMN-derived EVs together with stimulation via the TLR4 ligand, LPS, led to partial downregulation of cell maturation. This was visualized as a redundancy in surface marker expression (CD40, CD80, CD83, CD86, HLA-DP DQ DR), inhibition of cytokine release (TNF-α, IL-8, IL-10, and IL-12), as well as a reduction in T-cell proliferation. Taken together, this experiment provided strong evidence that PMN-derived EVs are capable of modifying both the function and maturation of monocyte-derived DCs [90]. In 2017, Slater et al. proposed a novel mechanism involving membrane-derived EVs being excreted by tissue infiltrating PMNs under inflammatory conditions, which provide active mediators that locally modulate cellular function. They demonstrated in vitro that PMN antimicrobial myeloperoxidase (MPO) can be mobilized to the PMN surface and subsequently released in association with PMN-derived EVs upon PMN activation and binding to IECs. Furthermore, PMN-EV-associated MPO enzymatic activity was stronger when compared to soluble protein, resulting in significant inhibition of wound closure following PMN-EV binding with IECs. These results have also provided strong evidence that EVs secreted by tissue-infiltrating PMNs form an efficient delivery system to locally modulate tissue function [91]. In a study from 2016, Butin-Israleli et al. presented an approach in which deposition of PMN membrane-derived EVs (PMN-EVs) onto IECs during transepithelial migration led to the loss of epithelial cadherins, which may increase PMN recruitment and cause epithelial disintegration. During transepithelial transit, PMN-EVs secreted by activated PMNs were capable of mediating intense effects on IEC integrity via a high level of enzymatically active MMP-9. Consequently isolated PMN-EVs can be efficiently linked to IEC monolayers and may trigger cleavage of desmoglein-2, but not E-cadherin, leading to disruption of IEC intercellular adhesions [92].

#### 2.2.3. EVs of Bacterial Origin

It is currently accepted that the occurrence of IBD can be attributed to an inappropriate immune response to normal commensal bacteria in individuals with a genetic predisposition [93]. In physiological conditions, the intestinal microbiome interacts with the host to maintain the epithelial gut barrier by influencing IEC metabolic activity and nutrient absorption. Furthermore, the intestinal microbiome acts to promote gut maturation and integrity and helps to maintain intestinal immune homeostasis [94]. In addition, the presence of bacteria exerts a strong protective effect on intestinal epithelium against harmful bacterial strains by inhibiting their colonization of the gut and by producing antimicrobial compounds. However, studies using culture-independent metagenomic methodology have confirmed that there are significant differences in the composition of microbiota in IBD patients when compared to healthy individuals [95]. Recent studies have demonstrated that in IBD patients, the number of bacteria with anti-inflammatory properties is lower, while the number of bacteria with pro-inflammatory properties is relatively higher [96,97]. As an example, in IBD, Bifidobacterium, Faecalibacterium, Odoribacter, and Lactobacillus, which have been shown to protect against mucosal inflammation via downregulation of inflammatory cytokines and upregulation of anti-inflammatory cytokines, are underrepresented, while *Enterobacteriaceae* are overrepresented [98,99,100]. Alterations in the composition of intestinal bacteria can influence intestinal homeostasis through various signaling pathways, mostly mediated via bacterial EVs (also known as OMVs). The significance of bacterial-derived EVs was demonstrated by Kang et al. using a C57BL/6 mice DSS-induced model of IBD. In this investigation, stool samples were evaluated by metagenomic sequencing using a bacterial common primer of 16S rDNA. Greater metagenomic changes in colitis occurred after a change in stool EV composition when compared to a change in bacterial composition alone. Furthermore, apart from other changes in bacterial EVs induced by DSS, pretreatment of *A. muciniphila*-derived EVs increased the production of IL-6 (a pro-inflammatory cytokine) from colon epithelial cells induced by *Escherichia coli* EVs [101]. One of most examined strains in the context of intestinal homeostasis is *Lactobacillus casei* BL23. A study by Rubio et al. examined the composition of bacterial membrane EVs and found a total of 103 proteins, of which 13 are exclusively present in the EVs. These EVs contained cell envelope-associated and secretory proteins, heat and cold shock proteins, metabolic enzymes, proteases, structural components of the ribosome, membrane transporters, cell wall-associated hydrolases, phage related proteins, as well as adhesion proteins (*Lactobacillus* p40 and p75), which indicates that EVs are key molecule carriers in the bacterial–gastrointestinal cell interface [102]. Although the spectrum of bacterial components involved in the proinflammatory response is not fully understood, the immunomodulatory effect of bacterial-derived EVs appears mostly dependent on its pro- and anti-inflammatory cytokine cargo or other components capable of stimulating cytokine synthesis pathways within the GI tract, such as ligands of pattern recognition receptors (PRR), lipoproteins, LPS, and peptidoglycan [103]. Bielaszewska et al. investigated the ability of EVs released during growth by enterohemorrhagic *Escherichia coli* (EHEC) O157 to induce production of pro-inflammatory cytokines in human IECs. They determined that EVs from EHEC O157 during growth can stimulate the production of IL-8 in IECs via the TLR5 and TLR4/MD-2 complex signaling pathway, followed by activation of NF-κB [104].

Other strains, such as *Bacteroides fragilis*, have been shown to exert protective properties against intestinal inflammation. In an animal model of colitis, Shen et al. showed that capsular polysaccharide (PSA) derived from *Bacteroides fragilis* OMVs may exert immunomodulatory effects and prevent mucosal inflammation. More specifically, they demonstrated that DCs can sense OMV-derived PSA via TLR2, which in turn leads to enhanced regulatory T-cell and anti-inflammatory cytokine production. Signaling induced by OMVs in DCs required growth arrest and DNA-damage-inducible protein (Gadd45α). Dendritic cells exposed to PSA-containing OMVs prevented experimental colitis, whereas Gadd45α(-/-) DCs were unable to induce regulatory T-cell responses or suppress pro-inflammatory cytokine production and mucosal pathological changes. This study revealed that OMVs derived from *Bacteroides fragilis* can trigger Treg development and IL-10 production through TLR2 signaling in DCs [105]. Other researchers, such as Ahmadi Badi et al., have also reported the effects of *Bacteroides fragilis*-derived OMVs on TLR2 and TLR4 gene expression, as well as the concentration of IFNγ, IL-10, and IL-4 in the Caco-2 cell line. They showed that TLR2 mRNA levels were not altered by *Bacteroides fragilis*-derived OMVs; however, these vesicles significantly altered TLR4 gene expression. Furthermore, they exerted a stimulatory effect on anti-inflammatory cytokines IL-4 and IL-10, while acting to decrease the concentration of IFNγ, a pro-inflammatory cytokine [106]. Taken together, it is clear that *Bacteroides fragilis*-derived OMVs play an important role in the enhancement of inflammatory responses.

Some of the most important receptors involved in host protection against bacterial infections and in the regulation of inflammatory responses are NOD1 and NOD2 cytosolic receptors. These receptors specifically recognize peptidoglycans present within the bacterial cell wall. Recent studies involving EVs from probiotic *Escherichia coli* Nissle 1917 and the commensal ECOR12 have shown they can activate NOD1 signaling pathways in IECs. Consequently, NOD1 silencing and RIP2 inhibition negated the EV-mediated activation of NF-κB and further expression of IL-6 and IL-8 [107]. The mucus layer is not an obstacle to crosstalk between microbiota and the host intestinal epithelium, and it was demonstrated that EVs can enter IECs via clathrin-dependent endocytosis [108]. Bacteria can also exert their immunomodulatory effects by secreting EVs containing polysaccharide A, which when delivered to intestinal DCs, can be sensed by TLR2. Moreover, some strains such as *Bacteroides thetaiotaomicron* can secrete hydrolytic enzyme-containing EVs which increase the digestion of gut microbiota, and they can share them with bacteria lacking these hydrolytic enzymes [105,109]. However, there are also bacterial strains that may aggravate IBD. A number of studies have indicated an increased load of bacteria that are attached to the intestinal mucus layer and may cleave the epithelial barrier. Among these bacteria, adherent invasive *Escherichia coli* (AIEC), *Fusobacterium varium*, and *Campylobacter jejuni* are the most studied [98]. Although the underlying mechanisms by which pathogens may invade IECs have not yet been fully elucidated, it was demonstrated that the disruption of cell-to-cell junctions might be caused by bacterial EV-induced proteolytic cleavage of E-cadherin and occludin, both of which are important in maintaining epithelial integrity [110].

## 3. Potential of EVs in Clinical Practice

EVs are of therapeutic interest because they are deregulated in both NEC and IBD, and they could be harnessed to deliver drugs to target cells. Since EVs are present in biological fluids, they may also serve as biomarkers in pathological conditions (Table 2).

### 3.1. Usage of EVs in Treatment

#### 3.1.1. Neonatal NEC

Several treatment strategies of NEC have been tested in animal models, and stem cell-derived EVs, as well as milk-derived EVs appear to be the most promising. It was shown in a rat model that intraperitoneal administration of BM-MSC EVs may exert a protective effect against NEC and decrease the severity of intestinal injury [42]. Further studies demonstrated that a reduction in NEC incidence could also be achieved by using EVs derived from other types of stem cells, such as amniotic-fluid-derived MSCs, BM-MSCs, amniotic-fluid-derived neural stem cells, and neonatal enteric neural stem cells [43]. Other studies, mainly focused on milk-derived EVs, have shown that these EVs are capable of significantly increasing intestinal villi height and crypt depth, thus improving the proliferation of intestinal epithelium and protecting against NEC [53,54,55]. Milk-derived EVs containing RNA, DNA, or proteins delivered to the infant intestine provide significant input into the proper understanding of the mechanisms by which certain drugs are delivered to the neonatal intestinal tract [117]. Of note, milk-derived EV cargo, including fragile particles such as miRNAs, are stable in the gut lumen environment even after simulated gastric/pancreatic digestion [118,119]. Stability of RNA particle (AF-488 siRNA) delivery along with its transepithelial transport was examined by fluorescence microscopy and fluorescence intensity measurements, which revealed that the encapsulation of siRNA in milk EVs resists harsh digestive processes, improving intestinal permeability and payload protection [120]. These results are in line with experiments performed using bovine milk and a human intestinal cell model. Bovine-milk-derived EVs increased goblet cell expression, enhanced mucin production, and increased goblet cell-associated markers, such as trefoil factor 3 and mucin 2 [117]. Although the above-mentioned results are promising, it must be pointed out that all EV-based therapies for NEC are still at the experimental stage.

#### 3.1.2. IBD

Targeting the origin of EVs, their cargo, and cellular uptake are potential therapeutic options. Current knowledge on the role of EVs in intercellular crosstalk and in IBD pathophysiology indicates that targeting the origin point of EVs or their specific cargo could be a promising therapeutic option. Components of the endosomal sorting complex required for transport (ESCRT) machinery and related molecules play essential roles in the biogenesis of EVs. Jackson et al. studied the effects of inhibiting the ESCRT-associated AAA+ ATPase VPS4 on EV release from cultured cells. They found that inhibition of VPS4 in HEK293 cells decreases the release of EV-associated proteins and miRNA, as well as the overall number of EVs [111]. Other studies have chosen a different path, by demonstrating in vitro that the control of EV secretion is possible via suppression of proteins involved in the docking and fusion of EVs with the plasma membrane [112]. Target cells may take up EVs through various pathways including clathrin-dependent endocytosis and clathrin-independent pathways, such as caveolin-mediated uptake or the CLIC/GLEEC pathway, macropinocytosis, phagocytosis, and lipid-raft-mediated internalization. Thus, current research strategies also involve numerous compounds, chemicals, and peptides that are capable of blocking EV uptake in different target cells [121]. Other considered directions include direct targeting of specific EV cargo components, such as inhibition or silencing of miR-21, which was found to be overexpressed in IBD patients [58,113].

EVs are potential drugs. There is also available data showing that EVs can be used as direct therapeutic agents. Under physiological conditions, IECs produce EVs with TGF-β1-dependent immunosuppressive activity. It was recently demonstrated that when these EVs were transferred into DSS-induced IBD mice, it led to decreased IBD severity through the induction of regulatory T-cells and immunosuppressive DCs. Furthermore, these EVs tended to localize in the intestinal tract and associated with EpCAM [63]. Recently, the usage of macrophage-derived EVs in the prevention of autoimmune diseases, including IBD, has evoked increasing interest. Yang et al. demonstrated that M2b macrophage-derived EVs exert protective effects on DSS-induced colitis, mainly mediated by the CC chemokine 1 (CCL1)/CCR8 axis [114]. Another possible approach is to mobilize molecules with epithelial repair properties such as ANXA1. It was shown that nanoparticles engineered using collagen IV (mimicking EVs) and containing the ANXA1 mimetic peptide, Ac2-26, can enhance recovery in DSS colitis mouse models and colonic biopsy-induced wounds [65]. This and other studies showing that EV cargo could be specially engineered provide a new perspective for the use of EVs as a potential means for drug delivery [122].

Based on the available literature, it is evident that immune-cell-derived EVs garner the most interest in the context of IBD therapy. Dendritic-cell-derived EVs are able to influence the progress of IBD via immunostimulatory or immunosuppressive effects. Wang et al. demonstrated that *S. japonicum* soluble egg antigen-treated EVs (SEA-treated DC EVs) may improve epithelial histological scores in acute DSS-induced colitis mice, thus preventing colon damage. The same experiment also revealed that in inflammatory conditions, SEA-treated DC exosomes can modulate the release of inflammatory cytokines [116]. Mesenchymal stem cells have been demonstrated to be useful for stem cell therapies in oncology, hematology, injury, as well as autoimmune disease [123]. Thus, stem-cell-derived EVs are considered a promising therapeutic option in IBD patients, which was demonstrated by Mao et al. In this investigation, MSC-derived EVs led to a significant reduction in several genetic inflammatory indices in DSS-induced IBD mice. More specifically, the expression of the IL-10 gene was increased, while that of TNF-α, IL-1β, IL-6, iNOS, and IL-7 genes was decreased in the colonic tissues and spleens of EV-treated animals. Decreased infiltration of macrophages into the colonic tissues was also observed. Furthermore, expression of IL-7 was greater in the colonic tissues of colitis than that of healthy subjects [124]. A strong paracrine effect was also described by other researchers who demonstrated in vitro that stimulation with TNF-α+IFN-γ MSCs-derived EVs may lead to a significant decrease in pro-inflammatory cytokine production [125] or attenuate oxidative stress and apoptosis in a TNBS-induced colitis model [126].

Finally, it is worth mentioning that besides *Schistosoma japonicum* [116], the therapeutic potential of other EVs of parasitic origin has also been studied. In several clinical trials, hookworm (*Nippostrongylus Brasiliensis*) infection has been demonstrated as a suppressive therapy against inflammation in the course of IBD and celiac disease [127]. In a study by Eichenberger et al., *Nippostrongylus Brasiliensis*-derived EVs were shown to exert a protective effect against intestinal inflammation in TNBS-induced colitis. This was reflected by a reduction in pro-inflammatory cytokines (IL-1β, IL-6, IL-17a, and IFNγ), decreased mucosal inflammatory infiltration, and reduced mucosal damage [115].

### 3.2. The Usage of EVs as Biomarkers

Currently, there are no EV-based tests available that might be used in a clinical setting as biomarkers for any autoimmune diseases. However, from the perspective of IBD, proteasome subunit alpha type 7 (PSMA7), ANXA1, and selected miRNAs, seem to be the most promising. In a study using liquid chromatography-mass spectrometry, PSMA7 enclosed in EVs from biological fluids (e.g., saliva), was shown to be associated with inflammation and the immune response [128,129]. Furthermore, it was clearly demonstrated that in active IBD, serum EV ANXA1 levels were elevated; thus, ANXA1 is considered to be a promising marker of mucosal inflammation [65]. Given that over 100 miRNAs have been demonstrated to be differentially expressed in IBD when compared to healthy individuals [130], these small particles have been extensively evaluated for their role in IBD diagnosis. Of these, most are differentially expressed in both UC and CD. The expression of some miRNAs, such as miR-16 or miR-21, was found to be upregulated in various tissues from both UC and CD patients when compared to normal controls [131]. Therefore, miRNA-containing EVs could serve not only as biomarkers in IBD diagnosis, but also for differentiation between UC and CD.

## 4. Conclusions

Cumulative evidence indicates that EVs are important players in cell-to-cell communication. They have the unique ability to transport membrane and cargo molecules, including proteins, lipids, DNA, and RNA between cells and to rapidly convey intercellular information. Independent of the type of cells, EV cargo selection is essential to affecting external conditions, and consequently, they have a great impact on intestinal homeostasis. Within the intestinal mucosa, both IECs and immune cells are routinely exposed to a great number of pathogens that may significantly perturb the epithelial barrier and immune function. To avoid exacerbation in bacterial abundance and/or overexpression of the inflammatory cascade within the intestinal wall, both host and bacteria constitute a specific EV signalization system, controlling each other on various levels and using a variety of different mechanisms. From the perspective of IBD, EVs are potentially vital contributors to autoimmunity and inflammation by carrying autoantigens, danger signals, cytokines, mediators, and tissue-degrading enzymes. Bacterial and maternal milk EVs are also crucial in the maturation of the neonatal GI tract as well as development of the immature GI immune system. Similarly, growing evidence indicates that those EVs may also be involved in complex signaling pathways controlling the magnitude, kinetics, and duration of various cellular responses in the neonatal GI tract. Therefore, novel methods using EVs may be useful as diagnostic tools. Moreover, they may significantly contribute to the treatment and prevention of neonatal intestinal diseases and IBD. Further studies are needed to confirm the beneficial effects of EVs before entering the clinical trial phase.

## Figures and Tables

**Table 2 cells-10-03202-t002:** Potential of EVs in clinical practice.

Potential Therapeutic Effect	Mechanism/Therapeutic Factor	Reference
Decreasing the release of EV and *miRNA*-related proteins	Inhibition of *VPS4* in *HEK293* cells	[111]
The control of EV secretion	Suppression of proteins involved in the docking and fusion of EVs with the plasma membrane	[112]
Direct targeting of specific EV cargo components	Inhibition or silencing of *miR-21*	[58,113]
Protective effects on DSS-induced colitis	*M2b* macrophage-derived EVs mainly mediated by the CC chemokine 1 *(CCL1)/CCR8* axis	[114]
Mobilization of molecules with epithelial repair properties	*ANXA1*	[65]
Reduction in pro-inflammatory cytokines, decreased mucosal inflammatory infiltration, reduced mucosal damage	*Nippostrongylus Brasiliensis*—derived EVs	[115]
Preventing colon damage	*Schistosoma japonicum* soluble egg antigen-treated EVs	[116]

EVs—extracellular vesicles, miRNAs—MicroRNAs, DSS—dextran sulfate sodium, ANXA1—Annexin-1.

## Data Availability

Not applicable.

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
