# Peer review of "An Update on the Role of Extracellular Vesicles in the Pathogenesis of Necrotizing Enterocolitis and Inflammatory Bowel Diseases"

_cells, 2021, doi:10.3390/cells10113202_

Round 1

Reviewer 1 Report

Excellent job in the revision. The manuscript is comprehensive, very informative, and reads very well. The only very minor change I would suggest is to add the word "partially" in the Abstract, in the sentence "The impact of bacteria on the host gut immune system is (partially) mediated via released extracellular vesicles (EVs). 

Author Response

  1.  

“Excellent job in the revision. The manuscript is comprehensive, very informative, and reads very well. The only very minor change I would suggest is to add the word "partially" in the Abstract, in the sentence "The impact of bacteria on the host gut immune system is (partially) mediated via released extracellular vesicles (EVs).”

Corrected as follows: The impact of bacteria on the host gut immune system is partially mediated via released extracellular vesicles (EVs).

Reviewer 2 Report

The manuscript reviews of the roles of extracellular vesicles of various origin in the normal function and pathophysiology of the gastrointestinal tract. 

The review should be substantially improved, especially in understanding this complex subject, by means of a graphic summary of the functions of EVs in the pathogenesis of IBD and their therapeutic use.

Author Response

Answers to Reviewer 2.

“The review should be substantially improved, especially in understanding this complex subject, by means of a graphic summary of the functions of EVs in the pathogenesis of IBD and their therapeutic use.”

As indicated by Reviewer 2, we attempted to improve the manuscript by adding Figure 1, Table1s 1 and 2, to enhance potential readers to understand functions of EVs in the pathogenesis of IBD and their therapeutic use.

Round 2

Reviewer 2 Report

The manuscript has been substantially improved, and the suggestions made have been properly addressed by the author. Thanks.

This manuscript is a resubmission of an earlier submission. The following is a list of the peer review reports and author responses from that submission.

Round 1

Reviewer 1 Report

The manuscript by Dr. Filip is an ambitious review of the roles of extracellular vesicles of various origin in the physiology and pathophysiology of the gastrointestinal tract. It is a relatively new field, which develops quickly and remains somewhat confusing when it comes to terminology, identification of effector molecules on and within the vesicles, their targets and distribution. As such, comprehensive reviews like this are in demand and can be very valuable for the field.

This review, while fundamentally valid in its intents and its content, could be improved in several areas to improve the language, flow, and ultimately reader’s comprehension:

Most importantly, the manuscript needs to be thoroughly reviewed by someone more proficient with English, ideally a native speaker, to correct the style and multiple grammatical and punctuation errors.

The review’s structure is not easy to follow. It would be easier to read and comprehend if the author would restructure it to follow some logical order. E.g., after introduction and EV classification, the major sections could be dedicated to EVs based on their origin: epithelial cells, immune cells, stem cells, bacteria. Within each of those sections, fundamental information and the relationship to gut maturation, NEC and IBD (as biomarkers and treatment options) could be discussed.

Under the role of EV derived from bacteria, the author should discuss the distention between vesicles derived from Gram-negative and Gram-positive bacteria. He should also describe the outer membrane vesicles (OMVs) from Bacteroides fragilis and their effects on the epithelial cells and in Treg differentiation and mucosal immune tolerance.

Whenever possible, cited reviews should be replaced by the original source papers.

In the beginning, the author seems to interchangeably use the term MV and CV (the latter acronym is not explained).

Reviewer 2 Report

The paper lacks a clear layout. There's so much unnecessary information included that the reader is frequently lost. I strongly recommend to reduce the amount of information by 50% and only include the parts that really deal with the role of EVs in NEC or IBD.

Also, the paper contains many grammar mistakes and requires extensive editing

Reviewer 3 Report

The review article “An update on the role of extracellular vesicles the pathogenesis of necrotising enterocolitis and inflammatory bowel diseases” by Rafal Filip summarizes some evidences regarding the current knowledge of extracellular vesicles biology and their roles in gut maturation and pathogenesis of NEC and IBD.

In the first place, the work needs an extensive and thorough review of the language used, prior to any type of consideration, both in terms of spelling and the editing of the text itself, apart from the structuring of the paragraphs, and the excessive use of abbreviations, many of which are unnecessary.

Several review articles have addressed the role of extracellular vesicles in the intestine, both in health and in pathological processes, some directly targeting IBD. As an example, here are some articles that address very similar topics:

Stanton BA. Extracellular Vesicles and Host-Pathogen Interactions: A Review of Inter-Kingdom Signaling by Small Noncoding RNA. Genes (Basel). 2021 Jun 30;12(7):1010. doi: 10.3390/genes12071010. PMID: 34208860; PMCID: PMC8303656.

Zhao L, Ye Y, Gu L, Jian Z, Stary CM, Xiong X. Extracellular vesicle-derived miRNA as a novel regulatory system for bi-directional communication in gut-brain-microbiota axis. J Transl Med. 2021 May 11;19(1):202. doi: 10.1186/s12967-021-02861-y. PMID: 33975607; PMCID: PMC8111782.

Cuesta CM, Guerri C, Ureña J, Pascual M. Role of Microbiota-Derived Extracellular Vesicles in Gut-Brain Communication. Int J Mol Sci. 2021 Apr 19;22(8):4235. doi: 10.3390/ijms22084235. PMID: 33921831; PMCID: PMC8073592.

Diaz-Garrido N, Cordero C, Olivo-Martinez Y, Badia J, Baldomà L. Cell-to-Cell Communication by Host-Released Extracellular Vesicles in the Gut: Implications in Health and Disease. Int J Mol Sci. 2021 Feb 23;22(4):2213. doi: 10.3390/ijms22042213. PMID: 33672304; PMCID: PMC7927122.

Ayyar KK, Moss AC. Exosomes in Intestinal Inflammation. Front Pharmacol. 2021 Jun 9;12:658505. doi: 10.3389/fphar.2021.658505. PMID: 34177577; PMCID: PMC8220320.

Haas-Neill S, Forsythe P. A Budding Relationship: Bacterial Extracellular Vesicles in the Microbiota-Gut-Brain Axis. Int J Mol Sci. 2020 Nov 24;21(23):8899. doi: 10.3390/ijms21238899. PMID: 33255332; PMCID: PMC7727686.

Wani S, Man Law IK, Pothoulakis C. Role and mechanisms of exosomal miRNAs in IBD pathophysiology. Am J Physiol Gastrointest Liver Physiol. 2020 Dec 1;319(6):G646-G654. doi: 10.1152/ajpgi.00295.2020. Epub 2020 Oct 7. PMID: 33026230; PMCID: PMC7792667.